# Experimental Validation of Real-Time Ski Jumping Tracking System Based on Wearable Sensors

**DOI:** 10.3390/s21237780

**Published:** 2021-11-23

**Authors:** Johannes Link, Sébastien Guillaume, Bjoern M. Eskofier

**Affiliations:** 1Machine Learning and Data Analytics Lab, Department Artificial Intelligence in Biomedical Engineering, Friedrich-Alexander-Universität Erlangen-Nürnberg (FAU), 91052 Erlangen, Germany; bjoern.eskofier@fau.de; 2Institute of Territorial Engineering, Haute Ecole d’Ingénierie et de Gestion du Canton de Vaud, 1400 Yverdon-les-Bains, Switzerland; sebastien.guillaume@heig-vd.ch

**Keywords:** sports, wearable sensors, ultra-wideband, inertial measurement unit, ski jumping, validation study, tracking

## Abstract

For sports scientists and coaches, its crucial to have reliable tracking systems to improve athletes. Therefore, this study aimed to examine the validity of a wearable real-time tracking system (WRRTS) for the quantification of ski jumping. The tracking system consists of wearable trackers attached to the ski bindings of the athletes and fixed antennas next to the jumping hill. To determine the accuracy and precision of the WRRTS, four athletes of the German A or B National Team performed 35 measured ski jumps. The WRRTS was used to measure the 3D positions and ski angles during the jump. The measurements are compared with camera measurements for the in-flight parameters and the official video distance for the jumping distance to assess their accuracy. We statistically evaluated the different methods using Bland–Altman plots. We thereby find a mean absolute error of 0.46 m for the jumping distance, 0.12 m for the in-flight positions, and 0.8°, and 3.4° for the camera projected pitch and V-style opening angle, respectively. We show the validity of the presented WRRTS to measure the investigated parameters. Thus, the system can be used as a tracking system during training and competitions for coaches and sports scientists. The real-time feature of the tracking system enables usage during live TV broadcasting.

## 1. Introduction

From the very beginning of ski jumping in the 19th century until now, the ski jumping technique has undergone various developments. This includes, amongst others, the Kongsberger Technique, in which the upper body is bent and the arms are extended over the head. Later, the arms were brought back next to the upper body. During all these changes, the skis remained parallel. The last major change was the introduction of the V-style in 1985, where the skis are not parallel but form a V [1,2]. Nowadays, this technique is used by almost all athletes. All these developments resulted in larger jumping distances. In the past, new jumping techniques were generally found by coincidence or trial and error. In contrast, today, researchers work hard to find the optimal flight technique [3,4,5,6] and study the biomechanics of ski jumping [7,8].

Therefore, sports scientists need reliable and automatic tracking systems. Various studies with different measurement systems have been conducted to investigate, amongst other things, the jumping distance [9], landing momentum [10], ground reaction forces [11], and ski position [12].

These studies incorporate the use of inertial measurement units (IMUs) [13,14], force insoles or differential Global Navigation Satellite System (dGNSS) [15]. The latter has the disadvantage of being obtrusive since antennas on the helmet and a backpack have to be attached to the ski jumpers. Additionally, camera-based tracking methods are used [5,7,16,17,18,19,20,21], which are unobtrusive but have the disadvantage of only covering a part of the jumping hill, or many cameras are needed and must be combined. Furthermore, the post-processing of video data has high computational costs and does not work during bad weather conditions.

The evaluated system uses ultra-wideband (UMB) radio communication and ranging in combination with IMUs to track the position of the athletes continuously during the flight. UWB has been used successfully in sport tracking applications in various sports. This includes indoor sports, such as ice hockey [22], and handball [23] and outdoor sports, such as soccer [24].

In previous studies, ski jumping parameters were obtained offline. For a later application during training sessions or TV broadcasting, a real-time system is beneficial and therefore required. Thus, e.g., the speed during the jump can be shown live during the TV broadcasting, or jumping trajectories can be compared directly using a 3D visualization. For a live application, wireless transmission of the obtained data is required. However, for the athletes and coaches, this also has several advantages. This includes easier handling, and the athletes are not interrupted in their focus, which is crucial in competitions and training due to the high risk associated with ski jumping.

Within this work, we investigate the accuracy of a wearable tracking system measuring several ski jumping parameters. The tracking system is unobtrusive, which is essential to be used in competitions so as not to distract the athletes during the crucial flight phase. The parameters include the jumping distance, in-flight positions, and in-flight orientation of the skis. All parameters are determined and transmitted in real-time. The wearable real-time tracking system (WRRTS) brings the lab to the field. In contrast to previously proposed systems, the evaluated system is real-time-capable. Another advantage of the investigated system over previous ones is the wireless data transmission, which results in no interaction with the athlete during training or competition and does not disturb their focus.

The main objective of this study is the systematic validation of a wearable real-time tracking system for ski jumping. Prior studies have only focused on single aspects of tracking ski jumps. The investigated system provides multiple measured parameters obtained and transmitted in real-time, making it very applicable for further research of the kinetics and kinematics of ski jumping. Therefore, the accuracy of the tracking system needs to be investigated so that sports scientists can derive valuable insights.

The remainder of the paper is structured as follows. Section 2 introduces the study procedure with the WRRTS as well as the reference measurement systems. Section 3 shows the results of the comparison of the WRRTS with its respective reference systems. In Section 4, the previously presented results of the validation study are discussed. Finally, in Section 5 the conclusions of this work are presented.

## 2. Materials and Methods

In this section, we introduce the procedure of the data acquisition and the different measuring methods with their respective setup at the ski jumping venue. Furthermore, the evaluation of the study data is presented.

### 2.1. Measurement Systems

The data of the jumps was acquired using the WRRTS, a camera system operated by ccc software GmbH, the total station tracking system QDaedalus [25,26], and the official video-based jumping distance measurement. All measurement systems are described in more detail in the following.

#### 2.1.1. Wearable Real-Time Tracking System (WRRTS)

We use the tracking system that was developed based on the previous work of Groh et al. [9,10] in combination with UWB-based tracking capabilities.

The WRRTS consists of two main components. The first main component of the tracking system comprises antennas with fixed positions along the jumping hill. Figure 1 shows the positions of the antennas along the ski jumping facility as well as the positions of the reference measurement devices.

The other main component is the mobile trackers that are attached to the skis of the athletes. In cooperation with the ski binding manufacturer Slatnar, the trackers were designed for easy attachment to the bindings to facilitate their use. An example of the attachment is shown in Figure 2.

The tracking system uses ultra-wideband radio technology and microelectromechanical inertial measurement units. This combination allows the continuous measurement of ski orientation, acceleration, velocity, and position of both skis during the complete jump in real-time. The internal update rate of the inertial measurement units is 1000 Hz, and that of the ultra-wideband radio technology as well as the transmitted output data is 20 Hz.

The measurements are acquired in a local Cartesian coordinate system centered at the middle of the edge of the jump-off platform, as shown in Figure 1.

Furthermore, a 3D scan of the landing hill is acquired using a total station. A total station is a theodolite with integrated distance measurement, thus measuring the vertical and horizontal angles and distance to an aimed point. This is used to obtain a mapping from the local Cartesian coordinate system to measurements with respect to the landing hill (e.g., height over ground, jumping distance).

#### 2.1.2. Official Distance Measurements

The jumping distance of the recorded jumps was determined with the official video-based measurement system that has been used in Fédération Internationale de Ski (FIS) competitions for over 25 years. The system operator was certified by FIS for video distance measurements. To determine the jumping distance, the system operator determines the first camera frame where both skis are flat on the landing hill. Through the camera calibration, the respective jumping distance is determined and rounded down to a resolution of 0.5 m. An example of the determination of the official jumping distance is shown in Figure 3.

#### 2.1.3. Camera Measurements (ccc Software GmbH)

In the field of sports software, ccc develops video analysis systems for training optimization and competition control and platforms for the storage of training and competition data [27].

ccc provided camera-based measurements of the ski jumps and was used as a reference for the validation study. To this end, multiple fixed cameras were installed next to the jump-off platform and next to the landing hill at 8 m, 18 m, 30 m, 45 m, and 60 m after the jump-off platform. Additionally, a pan–tilt–zoom (PTZ) camera was mounted at the upper end of the in-run. Due to a defective calibration, the deviation of the WRRTS in the *Y*-coordinate could not be investigated using the PTZ camera. The position of the PTZ camera was measured using the total station, enabling the projection of the V-opening-angle to be determined on the PTZ camera image plane. For the camera at 30 m, the calibration was faulty and therefore could not be used. Figure 1 shows the positions of all installed cameras.

For assessing the WRRTS measurements using the ccc videos, two different approaches were taken.

The first one was the projection of the 3D positions of the WRRTS into the 2D image plane using intrinsic and extrinsic camera calibration parameters. This approach does not provide a meaningful quantification of the measurement errors (only in pixels, not in meters). However, it provides an intuitive means for visually assessing the data quality since the WRRTS measurements can be seen directly in the image.

The second approach is the comparison of the video data with the WRRTS data in 3D. Since the jumper is only visible in one camera image at a time during a jump, full stereoscopic measurements are not possible. However, it is possible to associate every pixel coordinate in the camera image with a 3D vector that is formed by connecting the camera position (i.e., its optical center) with the corresponding point in the image plane. The deviation of the WRRTS positions from this 3D vector can then be determined. Based on this geometric model, comparisons of the ski orientation can also be performed. It should be noted that due to the two-dimensional nature of the camera images, only deviations in the image plane are captured by this method.

To compare the WRRTS position in 3D with the camera vectors, the trackers and skis are manually labeled in the videos. An example of the manual labeling of the skis and trackers for one of the cameras next to the landing hill is shown in Figure 4. Additionally, Figure 5 shows the manual labeling of the V-angle in the PTZ camera.

To compare the camera measurements, we determine the point of the interpolated WRRTS trajectory with the shortest distance to the camera vector. We use the instant of time of this point of the WRRTS trajectory for the position and angle comparison. This procedure is described in more detail in Section 3.

#### 2.1.4. Total Station Tracking (QDaedalus System)

Since the camera measurements did not provide full 3D positions of the skis, the 3D tracking system QDaedalus was additionally used for validating the three-dimensional position. QDaedalus is a measurement system developed at the Geodesy and Geodynamics Lab at the ETH Zurich and consists of a combination of total stations and CCD cameras that allow the accurate triangulation of objects in 3D. For this study, we used Leica TCA 1205 total stations in the QDaedalus system. The positions of the QDaedalus stations are shown in Figure 1. The raw position data measured with the QDaedalus system were filtered and interpolated using a least-squares collocation. The trajectories measured with QDaedalus were registered to the same coordinate system as the WRRTS measurements (see Figure 1) and compared to the WRRTS data regarding 3D positions.

### 2.2. Venue

The data acquisition took place at the hill size 100 ski jumping hill in Oberhof (Germany).

In contrast to the second day, foggy weather conditions impeded data acquisition on the first day of the acquisition. This affected some of the camera measurements during this day since the manual labeling requires a clear vision of the wearable tracker and skis. Additionally, the QDaedalus measurements were affected during this day since the measurement principles require inter-visibility between the total stations and the athletes.

In total, data of 35 jumps were collected over two days. Four athletes participated in the data acquisition (three male, one female). One of the athletes was part of the German A National Team and three of the German B National Team. Measurements of the jumps were acquired using the WRRTS and the reference measurement systems described below.

### 2.3. Evaluation

In this subsection, the synchronization of the WRRTS and the camera measurements, the procedure for the position, and angle comparison is described. Additionally, the statistical analysis is presented.

#### 2.3.1. Synchronization

The WRRTS and the camera system used for evaluation do not have a common synchronized time. Therefore, the measurements were synchronized in retrospect. Different synchronization procedures were used for the PTZ camera, and the cameras at the edge of the jumping hill.

To synchronize the PTZ camera with the WRRTS measurements, we applied a time offset correction. The internal time of the WRRTS was set to zero when passing the edge of the jump-off platform (origin of the local coordinate system). Therefore, by manually labeling the frame in the camera image where the tracker passed the edge of the jump-off platform, we determined the offset of the time of the PTZ camera. Using this offset, we synchronized both measurement systems for each jump individually.

To synchronize the lateral cameras at the side of the jumping hill with the WRRTS a different approach had to be used. Due to their limited field of view, the edge of the jump-off platform is not visible and, therefore, cannot be used as a reference point for synchronization. Instead, we take an alternative approach. First, we project the manually annotated pixel coordinates of the tracker into 3D using y=0. Then we linearly interpolate the measurements of the WRRTS. Then, the time offset that minimizes the distance between the interpolated WRRTS measurements and the line of sight between the camera position and the projected annotation is determined. This time offset is then used to align the WRRTS and camera measurements.

For the comparison of the *X*- and *Z*-coordinate with the camera measurements, we project the manual annotation in 3D using the *Y*-coordinate of the synchronized WRRTS measurement.

Since the WRRTS and QDaedalus measurements do not have a synchronized time base either, their temporal relation also had to be determined in retrospect. Therefore the trajectories of both systems are up-sampled via linear interpolation. The up-sampled trajectories are shifted in time to find the best temporal agreement. As the measure for agreement, the mean absolute error of the overlapping trajectories is used.

#### 2.3.2. Statistical Analysis

The results of the WRRTS and the respective reference system are statistically analyzed using different error measures. These are briefly explained in this subsection.

Bland Altman introduced a graphical approach to compare two measuring methods [28]. It consists of a scatter plot with the difference between paired measures against their mean. The mean of the two methods is used since both methods have a measurement error, and the true value is not known. Therefore the mean of the paired measures is the best estimate of the true value. Plotting the difference against the mean enables us to see if there is a dependency of the measurement error on the value of the measurement.

Additionally, the mean difference $\bar{d}$, as well as the upper and lower limit of agreement (LoA) is plotted as a horizontal line. The limits show the range of the difference in which 95% of the data are located. The limits of agreement are calculated as $\bar{d}\pm 1.96\;\cdot \;\sigma$, where σ is the standard deviation.

The assumption for the Bland–Altman plot is that the difference between the two methods is normally distributed. To test whether the assumption of a normal distribution is valid, we use a Kolmogorov–Smirnov test [29]. For a *p*-value ≥ 0.05, we accept the null hypothesis of a normal distribution. For a *p*-value < 0.05, the differences to a normal distribution are significant, and we test for other distributions to calculate the limits of agreement.

Apart from the mean, we also investigate the standard error of the mean (SEM). It is defined as
(1)$$\mathrm{SEM}=\frac{\sigma }{\sqrt{N}},$$
where *N* is the number of compared measurements and σ the standard deviation. The SEM gives an estimate of how far the mean of the sampled data differs from the mean of the whole population.

The deviation is summarized in terms of the mean absolute error (MAE), which is calculated as
(2)$$\mathrm{MAE}=\frac{\sum_{i=1}^{N}\left|x_{i,\mathrm{WRTTS}}-x_{i,\mathrm{REF}}\right|}{N},$$
where *N* is the number of compared measurements $x_{i,\mathrm{WRTTS}}$ is the *i*-th measurement obtained with the WRRTS (e.g., jumping distance), and $x_{i,\mathrm{REF}}$ is the *i*-th measurement obtained with the corresponding reference measurement system. The MAE is an easy-to-interpret measure combining the bias and precision of a distribution. Thus, we use it as the figure of merit for characterizing the accuracy of the WRRTS.

## 3. Results

### 3.1. Official Video Distance

The jumping distance determined with the WRRTS was compared to the official video distance.

The differences between measurement methods are visualized in Figure 6 using a Bland–Altman plot. The plot contains the jumping distance of 35 ski jumps. The jumping distances range from 70 m to 106 m. The bias of the WRRTS is 0.31 m with an SEM of ±0.08 m. The precision stands at 0.44 m which leads to upper and lower limits of agreement of −0.56 m and 1.17 m, respectively. Two data points exceed the limits of agreement. This corresponds to 5.7% of the data. The MAE amounts to 0.46 m. The data are distributed equally around the bias, with no dependency on the jumping distance.

### 3.2. Camera Measurements (ccc Software GmbH)

#### 3.2.1. Projection of 3D WRRTS Measurements into the 2D Image Plane

An example of the three-dimensional WRRTS position measurements projected into the image plane of a video frame can be seen in Figure 7. There, the cyan curve shows the projected trajectory of the tracker placed on the left ski of the athlete. The magenta curve is the projection of the right ski. Additionally, the coordinates for some of the measured samples are shown in the figure.

#### 3.2.2. Comparison of 3D WRRTS Position Measurements with 3D Camera Vectors

In this subsection, the in-flight positions, measured with the WRRTS, are compared to the camera measurements. Due to the missing calibrations, for the ccc camera placed at 30 m after the jump-off platform, these recorded videos could not be evaluated. Furthermore, for the PTZ camera, no calibration was available, so the *Y*-coordinate of the 3D positions could not be inspected with the camera measurements. Assuming a relatively small camera lens distortion, we nevertheless investigate the V-angles recorded by the PTZ camera.

Firstly, we have a closer look at the *X*-coordinate of the in-flight three-dimensional position. In Figure 8, the Bland–Altman plot visualizes the results. The data range from x=−1.6 m to x=58.4 m and consist of 741 paired measurements. The bias of the WRRTS is −0.039 m with an SEM of 0.002 m. The precision of the difference is 0.05 m. The resulting upper and lower limit of agreement are at −0.14 m, 0.07 m, respectively. The MAE condenses this to 0.04 m.

The difference does not follow a normal distribution or another basic distribution. The Bland–Altman plot shows that the bias of the difference varies with the *X*-coordinate. Additionally, as expected, the precision is worsening with larger *X*-coordinates. In more detail, the results for every camera individually are shown in Table 1.

The Bland–Altman plot that corresponds to the *Z*-coordinate is shown in Figure 9. The data range from z=−25.7 m to z=0.3 m and contain 741 paired measurements. The bias of the WRRTS is −0.071 m with an SEM of 0.003 m. The precision of the difference is 0.08 m, which leads to an upper and lower limit of agreement of 0.09 m and −0.23 m, respectively. The MAE of the difference stands at 0.08 m.

As the difference of the *X*-coordinate, the difference of the *Z*-coordinate does not follow a normal distribution or any other basic distribution. The precision of the difference increases for smaller *Z*-values, which means farther away from the jump-off platform. Furthermore, the bias of the WRRTS increases for decreasing *Z*-values. The influence of the camera position on the bias and precision of the *Z*-coordinate is shown in Table 1.

#### 3.2.3. Comparison of 3D WRRTS Position Measurements with 3D QDaedalus Measurements

In this subsection, the three-dimensional position during the jump determined with the WRRTS is compared with the results of the QDaedalus system. Due to the foggy weather on the first study day for this investigation, only jumps from the second study day are usable. This is because the jumpers could not be reliably seen and therefore not be tracked with the QDaedalus system.

This results in eight jumps investigated with a total of 618 data points. The data are visualized in Figure 10. Because we investigate three-dimensional data, a conventional Bland–Altman plot is not appropriate. Therefore, we plot the distance between the WRRTS position and the position of the QDaedalus system against the mean distance of both systems to the jump-off platform.

Since the distance between the points of both measurement systems is a projection of the distance vector, it is not normally distributed. We assume a beta distribution and test it with a Kolmogorov–Smirnov test. We therefore get a *p*-value of 0.38, which confirms our assumption. The right part of the plot shows a histogram of the projection of the difference between the two methods. Additionally, the fitted beta distribution is plotted.

The mean difference between both systems is 0.290 m with an SEM of ±0.007 m. Since the distance is a positive function, the mean is also the MAE. The precision of the difference is 0.18 m. The limit of agreement is not calculated as 1.96×σ but with the percentage point function of the beta distribution. We thereby get a limit of agreement of 0.64 m.

Looking at the scatter plot, the distance between the two methods tends to increase with the distance from the jump-off platform slightly.

#### 3.2.4. Comparison of 3D WRRTS Angle Measurements with 3D Camera Vectors

Lastly, we inspected the ski orientation during the flight phase. The difference between the angles in the ccc videos from the side of the landing hill and the WRRTS angles projected to the image plane are shown in Figure 11. The *X*-axis is thereby the mean of the angle to the unit vector in *X*-direction in three dimensions projected into the image plane. The data for the angle comparison consist of 662 paired measurements. The WRRTS has a bias of 0.26°, and the SEM stands at 0.04°. The precision of the difference is 1.1°. The MAE is 0.8°. The difference does not follow a normal distribution but a Student’s *t*-distribution. The Kolmogorov–Smirnov test for a Student’s *t*-distribution results in a *p*-value of 0.57. Using the respective percentage point function, the limits of agreement are determined to be −1.9° and 2.4 °.

Besides the cameras next to the landing hill, one PTZ camera is mounted at the start of the in-run. The videos recorded with this camera are evaluated and compared to the angle determined by the WRRTS. Due to the missing calibration of the PTZ camera, we can only determine the angles in the image plane but no positional information. The projected V-angle is compared at t=0.5,1.0,2.0s after crossing the edge of the jump-off platform.

Figure 12 visualizes the Bland–Altman plot for the V-angle determination. The data for the projected V-angle range from 41° to 76° and consist of 70 measurements. For some of the jumps, due to the fog on the first day of the study, no camera measurements could be taken. The WRRTS has a bias of 3.38° and an SEM of 0.24°. The standard deviation of the differences is 1.96°. Therefore, the limits of agreement stand at −0.47° and 7.23°.

The MAE of the differences of the projected V-angle amounts to 3.4°.

## 4. Discussion

### 4.1. Jumping Distance

The WRRTS determines the jumping distance in good agreement with the official video distance. Even though the accuracy of the WRRTS might not be sufficient for competitions, its usage is helpful for training since no manual operator is needed to determine the jumping distance individually. It is important to mention that the true jumping distance is unknown due to the rounding during the determination of the official video distance. This rounding down of the jumping distance of the official video distance introduces an MAE of 0.25 m compared to the unknown true value. Nevertheless, this MAE introduced by rounding cannot be subtracted from the MAE of the difference of the WRRTS and the official video distance as one might intuitively think.

The main advantage of the investigated system over previous studies [9] is the much higher accuracy. Groh et al. proposed an accuracy of 0.8±2.9 m compared to a camera system. Additionally, it has the advantage over the official jumping distance measurement of automating the process, so no FIS certified instructor is needed to label frames in the video manually. Therefore, the WRRTS is also faster than the official distance measurement.

### 4.2. Projection of 3D WRRTS Measurements into the 2D Image Plane

The projection of the 3D positions captured with the WRRTS showed good correspondence between the WRRTS measurements and the video material. The tracker positions of the jumpers in the videos are close to the projected WRRTS data, indicating that the WRRTS positions are plausible. Although this method does not allow a quantification of the measurement error in the local 3D coordinate system, it serves to provide a visual impression of the measurement quality in an intuitive way.

### 4.3. Comparison of 3D WRRTS Position Measurements with 3D Camera Vectors

The 3D in-flight positions of the WRRTS are compared to the QDaedalus and camera measurements. As seen in the corresponding Bland–Altman plots, the precision of the *X*- and *Z*-coordinate worsens during the jump, in comparison with the camera measurements. Since the bias also varies, a log transformation or investigation of the relative error does not lead to a difference following a normal distribution. Therefore, looking at the whole data range, the determined limits of agreement tend to be too far, which is also mentioned by Bland and Altman [28].

Calculating the MAE in 3D with only the camera measurements from the lateral cameras is not possible due to the missing calibration for the PTZ camera and thus no determined MAE of the *Y*-coordinate.

However, we calculate the MAE in 3D by combining the MAE of the *X*- and *Z*-coordinate determined with the camera measurements with the MAE of the *Y*-coordinate determined with the QDaedalus system (${\mathrm{MAE}}_{Y}=0.08\;m$). We thereby find an MAE in 3D of 0.12 m, which is more than a factor of two better than the MAE observed with the QDaedalus system. Since the precision of the differences depends on the precision of both measurement methods, the discrepancy of the MAE shows that the QDaedalus tracking introduces this difference in the MAE.

In comparison with a recent study, which uses dGNSS and achieves a better accuracy of smaller than 0.05 m [15], the advantage of the investigated system is that it is unobtrusive in contrast to the backpack and antennas at the helmet needed for the dGNSS.

Based on the 3D in-flight positions, the WRRTS also measures and transmits the in-flight velocity in real-time. Due to the accurately measured positions, we can assume that the velocity is also measured accurately, with its precision worsening with the distance to the jump-off platform.

### 4.4. Comparison of 3D WRRTS Angle Measurements with 3D Camera Vectors

The determination of the V-angle has a worse precision compared to the angle determined from the cameras next to the landing hill. One reason for this is that the V-angle is calculated using the orientation of both skis, which both introduce a measurement error. Apart from this, due to the missing calibration of the PTZ camera, no 3D vectors could be used to measure the V-angles. Instead, they were taken directly from the image. The angles measured by the WRRTS are projected onto the image plane, but effects such as radial distortion through the camera lens are not corrected. This may introduce a bias in the comparison of both methods. Another reason for the bias and worse precision may be the less accurate labeling of the PTZ camera images. This is introduced through the bending of the skis in conjunction with the jumper appearing relatively small in the image (see Figure 5). In contrast to this, in the cameras at the side of the landing hill, the jumper appears large, and thus the unbent part of the skis is identified more precisely (see Figure 4).

The investigated system clearly outperforms previous studies also based on IMUs, which achieve an angular precision of $-0.2\pm 4.8{}^{{}^{\circ}}$ for the lateral angle [13].

## 5. Conclusions

In this paper, we investigated the accuracy of a WRRTS for ski jumping. The system is based on UWB and IMU and capable of real-time data transmission. Overall, we see good correspondence for the investigated parameters. Table 2 summarizes the findings of this study.

The WRRTS and the respective reference system can be used interchangeably, accepting the determined bias and precision of their difference. The bias and precision of the difference is an estimation for the upper limit for the true unknown bias and precision of the WRRTS since the reference system also introduces a measurement error.

Looking at the possible applications of such a tracking system, we see great potential for the live application of the WRRTS during TV broadcasting, since information such as live-speed or trajectory comparison might be fascinating for the spectators. Furthermore, for the application during training, the WRRTS shows high potential to support coaches and sports scientists to improve further the technique of the athletes to jump even further.

## Figures and Tables

**Figure 1 sensors-21-07780-f001:**
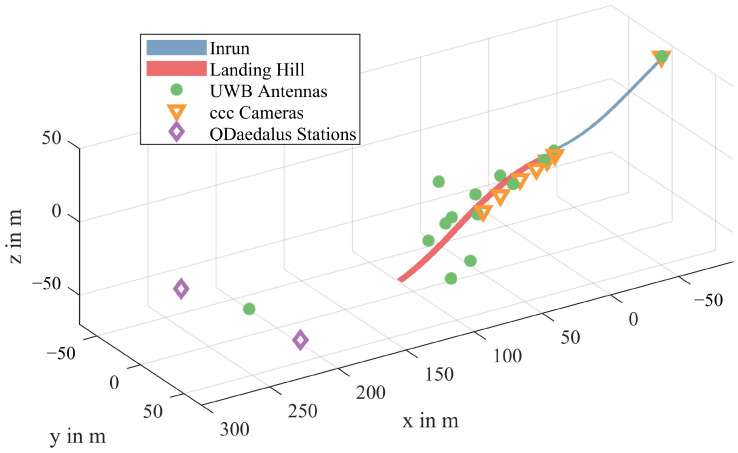
Definition of the local coordinate system with respect to the ski jumping hill. The origin of the coordinate system is the middle of the edge of the jump-off platform. With respect to the jumping direction, the axes are defined as: *X*-axis: horizontal, forward, *Y*-axis: horizontal, left, *Z*-axis: vertical, upward. Additionally, the setup at the ski jumping hill is presented. This includes the positions of the antennas of the WRRTS, the total stations of the QDaedalus system, the cameras, and the 3D models of the ski jumping in-run and landing hill.

**Figure 2 sensors-21-07780-f002:**
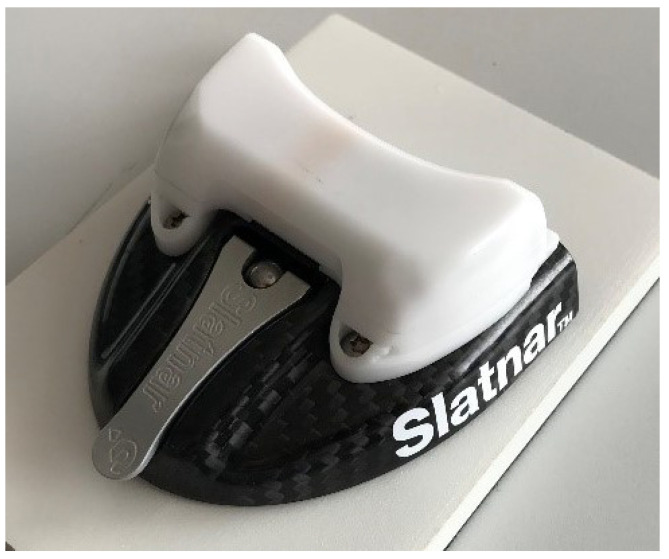
Attachment of the wearable tracker on top of the binding of the skis. The tracker is mounted on top of the regular ski jumping binding in front of the foot (tracker in white).

**Figure 3 sensors-21-07780-f003:**
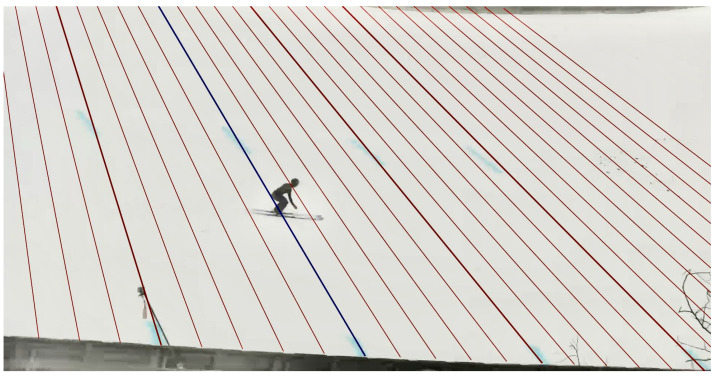
Example for the manual determination of the official video-based jumping distance. The red lines are the calibrated projection of the official jumping distance (to the jump-off platform). The blue line corresponds to the determined jumping distance.

**Figure 4 sensors-21-07780-f004:**
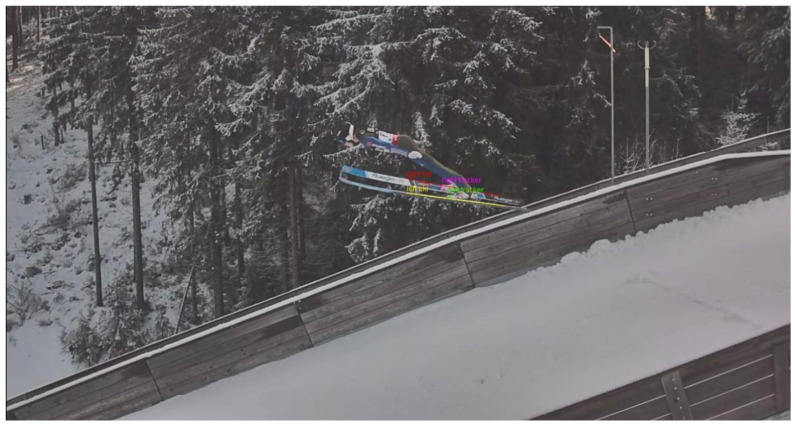
Example for the manual labeling of the tracker positions and skis for the cameras next to the jumping hill.

**Figure 5 sensors-21-07780-f005:**
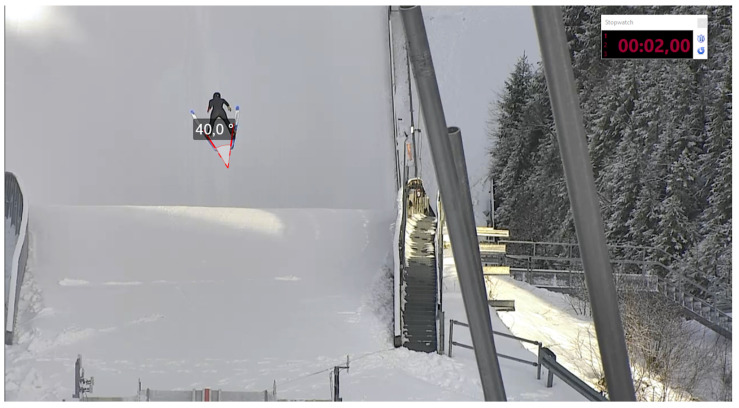
Example of the manual labeling of the V-angle 2 s after the crossing of the jump-off platform. The PTZ camera is mounted at the top of the in-run.

**Figure 6 sensors-21-07780-f006:**
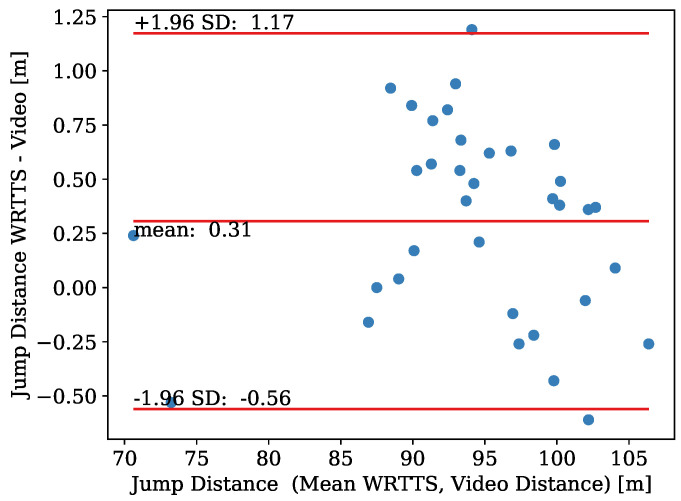
Bland–Altman plot for the comparison jumping distance determined with the WRRTS and the official video distance.

**Figure 7 sensors-21-07780-f007:**
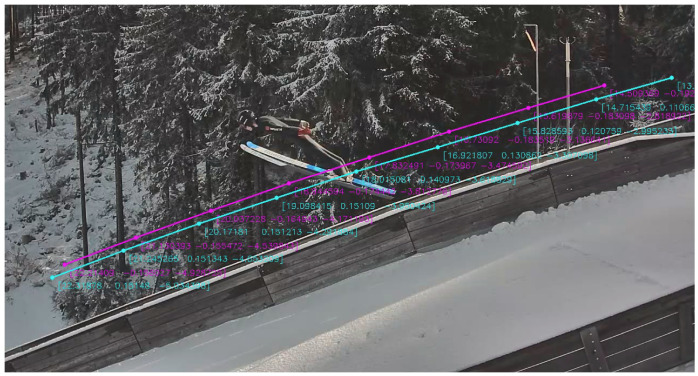
Projection of the WRRTS position data onto the video of the camera at 18 m after take-off. The cyan curve shows the trajectory of the WRRTS tracker placed on the binding of the left foot. The magenta curve shows the trajectory of the right tracker. The numbers indicate the corresponding 3D coordinates in meters. Coordinates as depicted in Figure 1.

**Figure 8 sensors-21-07780-f008:**
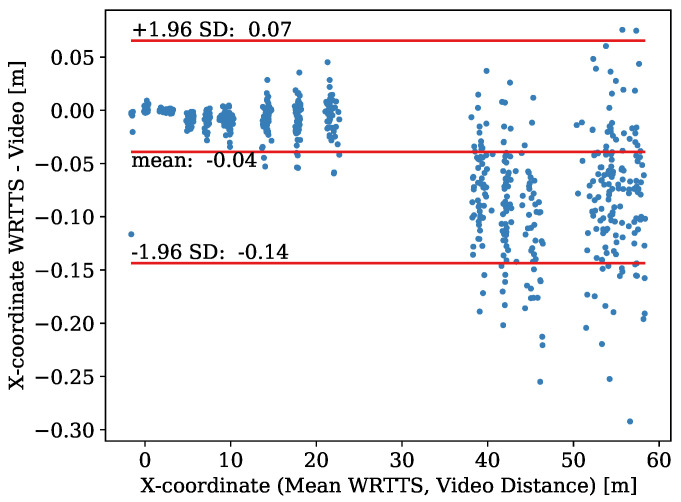
Bland–Altman plot for the comparison of the *X*-coordinate determined with the WRRTS and camera vectors.

**Figure 9 sensors-21-07780-f009:**
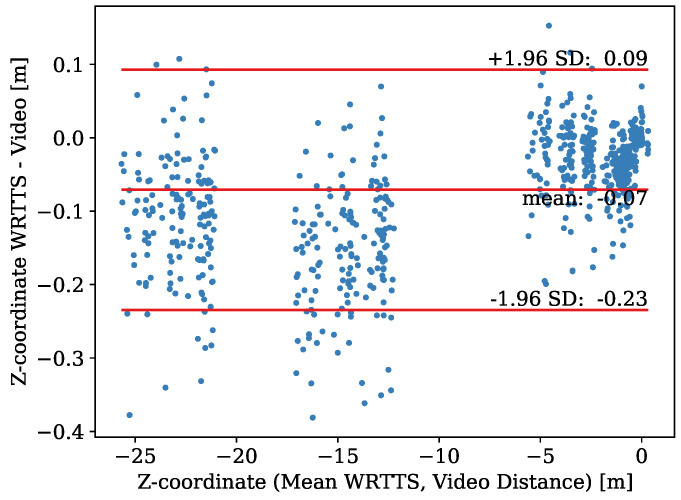
Bland–Altman plot for the comparison of the *Z*-coordinate determined with the WRRTS and camera vectors.

**Figure 10 sensors-21-07780-f010:**
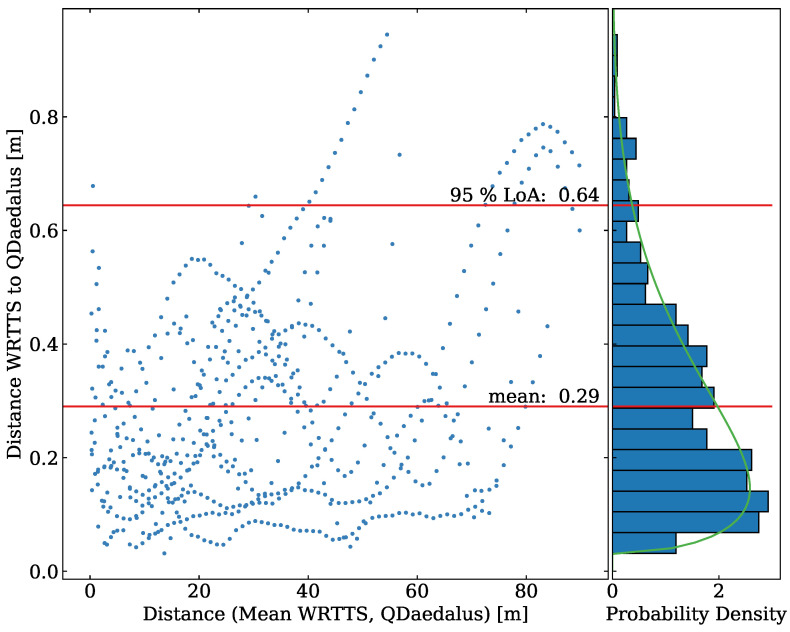
The left part shows a scatter plot for the comparison of the three-dimensional position measured with the WRRTS and QDaedalus tracking. The distance on the *X*-axis is the mean distance of WRRTS and QDaedalus to the jump-off platform. The histogram of the projection of the distance between both measurement methods is visualized in the right part. The fitted beta distribution is also plotted in the projection.

**Figure 11 sensors-21-07780-f011:**
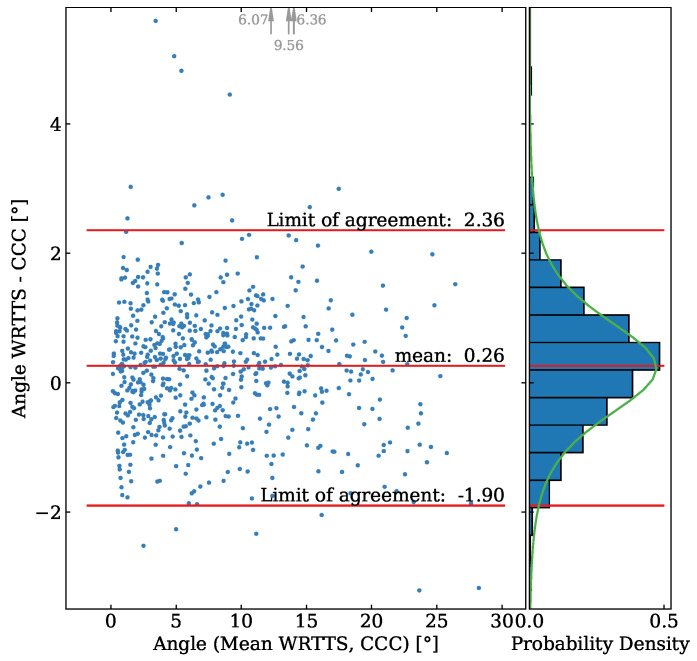
The left part shows the Bland–Altman plot for the comparison of the angle measured with the WRRTS and the angle determined in the camera images from the side along the jumping hill. The right part shows the projected histogram of the difference as well as the fitted Student’s *t*-distribution. The gray arrows and numbers at the top of the plot indicate three measured angles out of the plot’s range.

**Figure 12 sensors-21-07780-f012:**
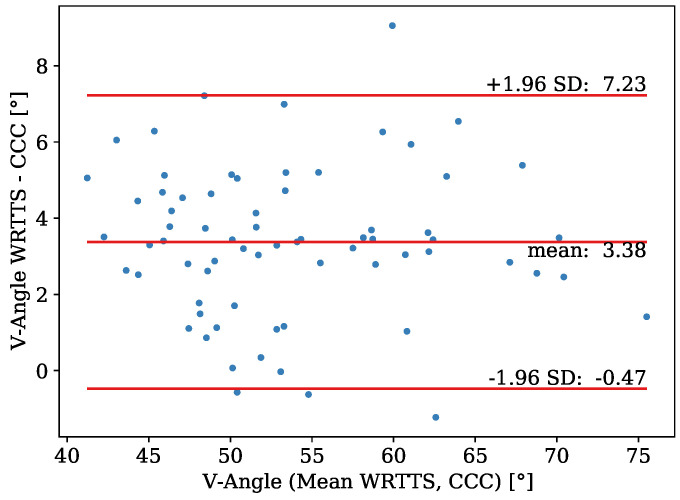
Bland–Altman plot for the comparison of the projected V-angle measured with the WRRTS and the PTZ camera.

**Table 1 sensors-21-07780-t001:** Bias, SEM, and precision of the *X*- and *Z*-coordinate determined with the WRRTS and camera vectors for every camera individually.

Coordinate	Metric	Camera Position				
		0 m	8 m	18 m	45 m	60 m
*X*	bias [m]	−0.001	−0.0086	−0.008	−0.088	−0.077
	SEM [m]	0.001	0.0005	0.001	0.004	0.005
	precision [m]	0.01	0.007	0.02	0.05	0.06
*Z*	bias [m]	−0.001	−0.048	−0.026	−0.151	−0.108
	SEM [m]	0.002	0.003	0.004	0.007	0.007
	precision [m]	0.02	0.03	0.06	0.08	0.08

**Table 2 sensors-21-07780-t002:** MAE between the WRRTS and the respective reference system for all parameters investigated in this study.

Parameter	Accuracy (MAE)
jumping distance	0.46 m
3D position	0.12 m
lateral angle	$0.8{}^{{}^{\circ}}$
V-angle	$3.4{}^{{}^{\circ}}$

## Data Availability

The data is available from the authors on reasonable request.

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
