# Peer review of "Experimental Validation of Real-Time Ski Jumping Tracking System Based on Wearable Sensors"

_sensors, 2021, doi:10.3390/s21237780_

Round 1
Reviewer 1 Report
This manuscript puts forward a real-time ski jumping tracking system and experimentally examined its performances. Bland-Altman plots are extensively used to test the accuracy and precision of their system in comparison with a video system. This paper can be accepted after minor revision.
- In the “Abstract”, line7, the authors mention that “velocity was measured”, but in the text, I cannot find data and discussion about velocity. Please address this.
- In the “Abstract”, line 12, a reference is cited to demonstrate the definition of “V-style opening angle”, as far as I understand. If this term has been widely accepted, there is no need to use a reference to denote.
Reviewer 2 Report
In this paper, authors develop a wearable system for real-time ski jumping tracking.
The authors do a good job at explaining their approach and showing various collected data, but they do not explain enough about previous related work and, most importantly, the-state-of-the-art that they are trying to outperform. Such systems have been used for other applications, so what contributions do the authors have from trying them for this application? From reading the paper, I understood that the most significant accuracy accomplishments were due to the devices used, not a custom technique developed by the authors. Therefore, I don't know if what the authors did could be considered a research contribution to the field rather than merely a pure engineering project. In addition, the English needs to be revised, especially with respect to punctuation.
Reviewer 3 Report
ABSTRACT: line 7: change "angels" for "angles".
INTRODUCTION: At the end add the objectives of the study .
MATERIAL AND METHOD AND RESULTS
line 113: provide any bibliographic reference
line 177-178: sentence introduction is not necessary
Figures: You have included a total of 14 figures. We consider it to be excessive for an article. Try to reduce the total number of them.
DISCUSSION: Add, in each of the points, a comparison of the advantages that this measurement system has provided with respect to other studies.
Round 2
Reviewer 2 Report
I think what the authors improved is enough for the paper to be accepted.